

# Response of Early Winter Haze Days in the North China Plain to Autumn Beaufort Sea Ice

Zhicong Yin [12], Yuyan Li[1] Huijun Wang[12]

[1]Key Laboratory of Meteorological Disaster, Ministry of Education / Joint International Research Laboratory of Climate and
Environment Change (ILCEC) / Collaborative Innovation Center on Forecast and Evaluation of Meteorological Disasters
(CIC-FEMD), Nanjing University of Information Science & Technology, Nanjing 210044, China

[2]Nansen-Zhu International Research Centre, Institute of Atmospheric Physics, Chinese Academy of Sciences, Beijing, China

*Correspondence to*: Yuyan Li (yyan370@163.com)

**Abstract.** Recently, early winter haze pollution in the North China Plain has been serious and disastrous, dramatically damaging human health and the social economy. In this study, we emphasized the close connection between early winter haze days in the North China Plain and the September-October sea ice in the west of the Beaufort Sea (R=0.51). Due to efficient radiative cooling, the responses of atmospheric circulations partially manifested as reductions of surface wind speed over the Beaufort Sea and Gulf of Alaska, resulting in a warmer sea surface in the subsequent November. The sea surface temperature anomalies over the Bering Sea and Gulf of Alaska acted as a bridge. The warmer sea surface efficiently heated the above air and led to suitable atmospheric backgrounds to enhance the potential of haze weather (e.g., a weaker East Asia jet stream and a Rossby wave-like train propagated from North China and the Japan Sea, through the Bering Sea and Gulf of Alaska, to the Cordillera Mountains). Near the surface, the weakening sea level pressure gradient stimulated anomalous southerlies over the coastal area of China and brought about a calm and moist environment for haze formation. The thermal inversion was also enhanced to restrict the underswing of clear and dry upper air. Thus, the horizontal and vertical dispersion were both limited, and the fine particles were apt to accumulate and cause haze pollution.

**Keywords:** Haze, Pollution, Aerosol, Sea ice, Arctic, Climate change

## 1. Introduction

In February 2018, the highest temperature near the Arctic region was above the freezing point (Jason, 2018), raising tremendous concerns from global climate change scientists. During the past few years, the increase of surface air temperature has been distinctly amplified in the Arctic area (i.e., the Arctic Amplification feature) and approximately twice as large as the average increase in global warming (Zhou, 2017). Recently, the Arctic sea ice (ASI) decreased rapidly due to the Arctic Amplification and reached a record low in September 2012 (Gao et al., 2015). The loss of ASI changed the reflection of solar radiation and the exchange of energy and fresh water, which could remotely connect with the climate in the Northern Hemisphere, especially the winter climate variability in Eurasia (Liu et al., 2007; Wang and Liu, 2016). As an



efficient external driver in the high latitudes, the decreased ASI over the Barents–Kara Seas in late autumn stimulated a

planetary-scale Rossby wave train in early winter (Honda et al., 2009; Kim et al., 2014) and transported its impacts to

Eurasia. The variation of the autumn ASI had significant impacts on the East Asian jet stream and the East Asian trough (Li

and Wang, 2013) as well as the winter Arctic Oscillation (Li and Wang, 2012; Li et al., 2015) and the East Asian winter

monsoon (Li and Wang, 2014; Li et al., 2014). Since 2000, the snowfall in Siberia has been enhanced, which is probably

related to the increased moisture flux from the Arctic (Cohen et al., 2012; Li and Wang, 2013). Liu et al (2012) illustrated

that the decrease of autumn ASI resulted in more blocking patterns and water vapor, which was a benefit for heavy snowfall

in Europe during early winter and in the United States during winter. Furthermore, under the positive Pacific decadal

oscillation phase, the autumn ASI reduction contributed to the subseasonal variability of surface air temperature in the East

Asian winter (Xu et al., 2018; He, 2015). The dust and sandstorm over North China, types of weather that are sensitive to

wind, also showed close relationships with the variation of ASI after the mid-1990s (Fan et al., 2017). The sea ice over the

Barents–Kara Seas induced dust-related atmospheric circulations (e.g., a strengthened East Asian jet, increased cyclogenesis,

and greater atmospheric thermal instability).

Haze, also being sensitive to wind, frequently occurred under calm and static weather conditions, i.e., small surface

winds and strong thermal inversion (Yin et al., 2015; Ding and Liu, 2014; Cai et al., 2017; Gao and Chen, 2017). For the

long-term trend of haze, human activities are the recognized and fundamental driver (Li et al., 2018; Yang et al 2016), but the

rapid ASI decline also contributed to the trend of haze pollution in the North China Plain after 2000 (Wang and Chen 2017).

For the interannual to interdecadal variations, the impacts of ASI on the haze in the east of China were emphasized by

observational analyses (Wang et al., 2015) and numerical studies (Li et al. 2017). From 1979–2012, the ASI loss led to a

northward shift of the East Asia jet stream and weak East Asian winter monsoons, indicating a strongly negative correlation

with the haze in the east of China (Wang et al. 2015). However, the first mode of the Empirical Orthogonal Function (EOF)

in Yin and Wang (2016a) presented different variations of haze days in the south and north of the Yangtze River. The positive

relationship between the autumn sea ice in the Beaufort Sea and winter haze days was briefly revealed without sufficient

physical explanations but contributed to the prediction of winter haze days (Yin and Wang 2016b, 2017b). The early winter

(December-January) haze days also varied differently with the February haze days (figure omitted), indicating a variant

modulating mechanism from the climate drivers. Thus, an open question still existed, i.e., the connections between Beaufort

Sea ice (BSI) and early winter haze pollution in the North China Plain (NCP: 34–42ºN, 114-120ºE) and the associated

physical mechanisms.

## 2.   Datasets and methods

The monthly sea ice concentrations (1 º×1 º) were downloaded from the Met Office Hadley Center (Rayner et al. 2003),





which is widely used in the sea ice-related analysis. The haze days were mainly calculated with the 6-hr observed visibility and relative humidity. The observed relative humidity, visibility, wind speed, and weather phenomena data used here were collected and controlled by the National Meteorological Information Center, China Meteorological Administration. The computing method of haze days was in accordance with Yin et al. (2017). The hourly $PM_{2.5}$ concentration data were provided by the Ministry of Environmental Protection of China. The daily maximum $PM_{2.5}$ was the maximum value obtained over

24-hour measurements. The $1°×1°$ ERA-Interim data used here included the geopotential height (Z), zonal and meridional wind, specific humidity, vertical velocity, air temperature at different pressure levels, sea level pressure (SLP), boundary layer height and surface air temperature data (Dee et al. 2011). The monthly mean sea surface temperature (SST) datasets, with a horizontal resolution of $1°×1°$, were also derived from the website of ERA-Interim (Dee et al. 2011). The $2.5°×2.5°$ monthly reanalysis heat fluxes (i.e., the sensible heat net flux and the latent heat net flux) were available on the website of

the National Center for Environmental Prediction and the National Center for Atmospheric Research (Kalnay et al. 1996).

### 3. Variation of the early winter haze

In most of the observational sites in the east of China, the number of haze days in December and January (HDJ) accounted for more than 70% of the total winter haze days (Figure 1), indicating that the haze pollution in the early winter was more serious than that in February. Yin et al (2018) also illustrated that the inter-annual variation of haze days in

February was different from that in the early winter. Thus, it is necessary to analyze the features of haze pollution in the early winter and associated climate drivers. The observational HDJ were decomposed by the EOF method and the variation contribution of the first and second modes were 33% and 14%, respectively. In the first mode, the HDJ in the south and north of the Yangtze River varied differently (Figure 1) and should have a distinguishing relationship with the autumn ASI. In this study, we focused on the HDJ in the NCP region ($HDJ_{NCP}$) and its connection with the autumn ASI.

The $HDJ_{NCP}$ varied approximately 30 days from 1979 to 1992 and decreased slightly from 1993 to 2007. Since 2008, the inter-annual variation of $HDJ_{NCP}$ has become more evident. The minimum $HDJ_{NCP}$ occurred in 2010, which was 17.5 days. Afterwards, the $HDJ_{NCP}$ increased dramatically and persistently, reaching a maximum (i.e., 42.7 days) in 2015. The mass concentration of $PM_{2.5}$ is an important indicator of haze pollution. The daily maximum of area-mean $PM_{2.5}$ in 2015 is shown in Figure 2b and was above 100 μg/m³. The synoptic processes of haze were relatively weaker in January 2016 than

those in December but still exceeded the threshold of pollution in China. On 23 December, the most disastrous haze occurred, and the area-mean $PM_{2.5}$ concentration approached 500 μg/m³, indicating quite poor air quality and a serious health risk.





### 4.    Connection with ASI and associated physical mechanisms

As illustrated by Wang et al (2015), the autumn ASI significantly and negatively affected the haze pollution in the east of China by modulating the large-scale atmospheric circulations and local meteorological conditions. Furthermore, the

opposite pattern of haze days in the east of China was revealed in Figure 1. To confirm the response of $HDJ_{NCP}$ to the autumn sea ice, the correlation coefficients between the $HDJ_{NCP}$ and the September-October sea ice were assessed and are shown in Figure 3. A positive correlation was found from the East Siberian Sea to the Beaufort Sea. In this broad region, the significantly correlated area was intensively located over the west of the Beaufort Sea. Thus, the area-averaged September-October sea ice area over the west of the Beaufort Sea (73-80 °N, 146-178 °W) was calculated and denoted as the

BSISO index, whose correlation coefficient with $HDJ_{NCP}$ was 0.51 (above the 99% confidence level) after removing the linear trend. This apparent positive relationship indicates that the efficient accumulation of the preceding autumn sea ice over the west of the Beaufort Sea significantly intensified the early winter haze pollution over the NCP area. To confirm this connection, the year-to-year change of the sea ice concentration was examined (Figure 4). From 1979 to 2015, there were seven years with significantly negative BSISO (i.e., BSISO<–0.8×its standard deviation), and ten years with significantly

positive BSISO (i.e., BSISO>0.8×its standard deviation). During 65% of these years, the significant BSISO anomalies induced $HDJ_{NCP}$ anomalies with the same mathematical sign. There were no significantly opposite responses of $HDJ_{NCP}$ ($|HDJ_{NCP}|$> 0.8×its standard deviation) to the BSISO anomalies. Furthermore, the relationship seemed to be enhanced after the mid-1990s. The same mathematical signs of the anomalies appeared more frequently, and the intensity of the responses was also amplified.

The heavy sea ice, with high albedo, can efficiently reflect solar radiation and restore more fresh water, which could influence the local and adjacent SST. The correlation coefficients between BSISO and the simultaneous and subsequent SST were computed (Figure 5). Because of efficient reflections of the solar radiation, the locally negative SST anomalies, located near the west of the Beaufort Sea (70–81 °N, 166 °E–138 °W), were induced by the heavy BSISO in October. In the following two months, these negative SST anomalies could not be sustained, i.e., these anomalous responses disappeared in November.

However, the induced positive SST anomalies in the Bering Sea (49–60 °N, 165–180 °W) and the Gulf of Alaska (40–52 °N, 130–165 °W) appeared in October and were persistently enhanced in November and December. These three significantly correlated SSTs, located near the west of the Beaufort Sea (WB), over the Bering Sea (BS) and the Gulf of Alaska (GA), were defined as $SST_{WB}$, $SST_{BS}$ and $SST_{GA}$, respectively. The correlation coefficients between these three indices were enumerated in Table 1 to present the change of the correlationship with SST. Over time, the linkage between BSISO and

local SST (i.e., $SST_{WB}$) rapidly receded. To confirm the role of the local SST on the $HDJ_{NCP}$, the correlation coefficient between $HDJ_{NCP}$ and $SST_{WB}$ was –0.30 in October (exceeding the 95% confidence level), and –0.05 and –0.10 in the following November and December (insignificant). However, the correlation coefficients between the $SST_{BS}$ ($SST_{GA}$) and



BSISO were persistent and even became enhanced in November and December (Table 1). We speculated that the November $SST_{BS}$ and $SST_{GA}$ was the junction between the BSISO and $HDJ_{NCP}$.

According to the numerical results illustrated by Deser and Tomas (2007), the responses of atmospheric circulations to sea ice anomalies were initially baroclinic in the first 5–10 days and progressively became more barotropic and increased in both spatial extent and magnitude within 2 months. In September and October, due to the radiative cooling of the heavy BSISO, the baroclinic responses of the atmospheric circulations manifested mainly as anomalous cyclonic circulation in the upper troposphere (Figure 6a). There were also weak anti-cyclonic responses in the Bering Sea and Gulf of Alaska. In the

subsequent November, the extent of these induced cyclonic and anti-cyclonic anomalies increased, especially the anti-cyclonic circulations over the Bering Sea and Gulf of Alaska (Figure 6b). The barotropic structure of the atmospheric responses became more obvious, i.e., there were also cyclonic and anti-cyclonic circulations on both sides of the Beaufort Sea, near the surface (Figure 6d). In addition, there were also positive SLP anomalies near the Aleutian Islands, indicating a weak Aleutian Low. Near the surface, a significant anomalous southerly was induced between cyclonic and anti-cyclonic

circulations, and an anomalous east wind was excited in the south of the anti-cyclonic circulation (Figure 6d). Overlapping with the climate mean state, the surface wind speeds over the RS1 (41–54 °N,140–165 °W) and RS2 (70–76 °N,140 °–170 °W) regions were significantly receded (Figure 7). The area-average surface wind speed was then calculated and denoted as $WSPD_{RS1}$ and $WSPD_{RS2}$ to examine its impacts on the simultaneous SST. In November, the climatological northeasterly through the Bering Strait transported cold seawater from the Arctic to the Bering Sea and resulted in a lower SST. The

correlation coefficient between $WSPD_{RS2}$ and SST is shown in Figure 8 and was significantly negative in the Bering Sea. The driver of the cold seawater transportations, i.e., the surface wind, decreased and led to warmer $SST_{BS}$ in November. Another reduction of surface wind speed, i.e., $WSPD_{RS1}$, indicated the weakening of the west surface wind and accompanying subdued evaporation near the sea surface. This RS1 region was located consistently with the warmer Gulf of Alaska. The correlation coefficients between the $WSPD_{RS1}$ and $SST_{GA}$ were significantly negative, indicating that the

reduction of $WSPD_{RS1}$ resulted in a warmer sea surface over the Gulf of Alaska (Figure 9a). Due to the weakening of the water evaporation, the latent heat release slowed down both in the Bering Sea and the Gulf of Alaska, which conserved more thermal energy in the sea surface (Figure 9b). In addition, the upper anti-cyclonic circulations, with clear sky, facilitated more shortwave solar radiation onto the sea surface. The absorbed and stored thermal energy, which was connected with the heavy BSISO, heated the sea surface over the Gulf of Alaska in November, i.e., positive $SST_{GA}$ anomalies. Both of the

$SST_{GA}$ and $SST_{BS}$ were significantly influenced by the BSISO and synchronously changed (figure omitted). Thus, the $SST_{BS}$ and $SST_{GA}$ were integrated as $SST_{BA}$ to analyze their corporate impacts on the $HDJ_{NCP}$. The variations of November $SST_{BA}$ and the BSISO were strongly consistent, especially after 2000 (Figure 10). As presented in Table 1, from October, the $SST_{BA}$ began to significantly connect with the BSISO. Over time, this connection persisted and strengthened. The correlation



coefficient between November (December) SST$_{BA}$ and the BSISO was 0.43 (0.48), exceeding the 99% significance test.

Statistically, the November SST$_{BA}$ was significantly correlated with the HDJ$_{NCP}$ (i.e., the correlation coefficient was

0.61 and above the 99% confidence level), showing strong impacts on the early winter haze pollution over the NCP region.

To reveal the physical processes, the associated atmospheric circulations and local meteorological conditions were diagnosed

in Figure 11–15. The warmer sea surface efficiently heated the above air and resulted in ascending motion from the Gulf of

Alaska to the Aleutian Islands, which could extend to the atmosphere at 200 hPa (Figure 11). Furthermore, significant

accompanying descending motions at 200 hPa were stimulated from the Sea of Okhotsk to the Hawaiian Islands (Figure 11a).

Near the surface, there was also a sinking motion over the Hawaiian Islands (Figure 11b). On the mid-troposphere, the

significantly negative Z500 anomalies, i.e., cyclonic circulations, were exerted above the warmer Bering Sea and Gulf of

Alaska (Figure 12b). The responses of the December-January atmospheric circulations to the warmer November SST$_{BA}$

showed deeply barotropic structures. There were also significant cyclonic circulations in the lower troposphere (Figure 12c)

and near the surface (Figure 12d). At 500 hPa, there were significant anti-cyclonic anomalies located on both sides of the

cyclonic circulations, i.e., over North China, the Japan Sea, and the Cordillera Mts. Thus, a Rossby wave-like train was

induced by the SST$_{BA}$, which propagated from North China and the Japan Sea, through the Bering Sea and Gulf of Alaska, to

the Cordillera Mts. This "+–+" pattern could also be recognized in the lower (Figure 12c) and upper (Figure 12a)

troposphere. The anti-cyclonic circulations over North China and the Japan Sea were recognized as the key atmospheric

system to influence the haze pollution in the NCP area (Yin and Wang, 2016a; Yin et al., 2017). To confirm the linkage

between this Rossby wave-like train and SST$_{BA}$, the area-averaged Z500 in three centers ([88 E–115 E, 30–50 N],

[150 E–160 W, 45–60 N], [115 W–130 W, 50–60 N]) were calculated and are shown in Figure 13. The correlation

coefficients between the three centers, from west to east, with SST$_{BA}$ were 0.47, –0.46, and 0.37, all above the 95%

confidence level. Due to the change of the pressure gradient, there were positive zonal westerlies from Lake Baikal to the

Hawaiian Islands and negative westerly anomalies from East China to the west subtropical Pacific (Figure 12a). Therefore,

zonal west winds prevailed in the mid-high latitude, and the meridionality of the atmosphere was reduced. The East Asia jet

stream was weakened by anomalous easterlies and shifted northwards, indicating the decrease of the southward cold air

activities. In addition, the southern section of the East Asia major trough, which reached the NCP area and guided cold air

southward, was truncated by the anomalous anti-cyclonic circulations (Figure 12b). In contrast, due to the cyclonic

anomalies over the Aleutian area, the northern section of the East Asia major trough was enhanced but moved eastward.

These large-scale anomalous atmospheric circulations could provide a suitable background for the enhancement of the

potential of the haze weather (Yin and Wang, 2017a).

          Near the surface, because of the SST$_{BA}$ heating, the Aleutian Low moved eastward and was enhanced over the Bering

Sea and Gulf of Alaska. Consistent with the barotropic anomalies in the above air, there were positive SLP anomalies from



Northeast China to the west Pacific (Figure 12d). That is, a north-south seesaw over the North Pacific was discerned clearly,

similar to the anomalous North Pacific Oscillation (NPO) pattern (Rogers, 1981). The difference of SLP ([140 E–170 W,

20–30 N] minus [165 E–155 W, 48–65 N]) was calculated to quantify this seesaw pattern, whose correlation coefficients

with the BSISO, $SST_{BA}$ and $HDJ_{NCP}$ were 0.33, 0.64 and 0.61, respectively. Bounded by the east of China, the south positive

center of NPO and the negative anomalies occupied the west Pacific and Eurasia, respectively. The drivers of the East Asian

winter monsoon, i.e., the pressure gradient between the continent and the ocean, became weak, indicating the limitation of

cold air and ventilation conditions. The induced southerly anomalies were located over the coastal area of China and

transported moisture to the NCP area (Figure 14a), providing moist air for haze formation. In winter, the anomalous south

winds also weakened the prevailing northerly and reduced the invasion of cold air. The humid atmosphere was conducive to

the hygroscopic growth of pollutant particles, which reduced the visibility rapidly and structured stable weather conditions.

The surface wind speed, indicating the horizontal dispersion capacity of the atmosphere, also subsided. In addition, the

shallow thermal inversion layer or the boundary layer limited the upward dispersion of the pollutant particles. As shown in

Figure 14b, the intensity of the thermal inversion over the NCP area was significantly heightened, while the boundary layer

significantly declined. Generally, the air on the high altitude was relatively dry and clean. The sinking of the upper air to the

surface was an important approach in dispersing the surface pollution (Sun et al., 2017). Instead, the upward motion above

the boundary layer resisted the breaking of the thermal layer and was in favor of haze occurrence (Figure 15a). The

associated anomalous descending flow was blocked in the north of 46ºN, which was consistent with the location of the

northward cold air activities (Figure 15a). Influenced by the warmer $SST_{BA}$, there were anomalous ascending motions over

the NCP area. Thus, the weakened downward transportation of momentum was not sufficient to enhance the winds near the

surface and break the thermal inversion layer. Therefore, the clear, dry and cold air was difficult to transport to the surface,

indicating the failure of the blowing wind. Under poor ventilation conditions, i.e., the horizontal and vertical dispersion was

limited, the fine particles were apt to accumulate and cause haze pollution. Combined with favorable moisture conditions,

the haze exacerbated rapidly and perniciously.

## 5.    Conclusions and discussions

In the sub-seasonal scale, the haze weather in early winter occurred more frequently and varied differently from that in

February. In this study, the close relationship between early winter haze days in the NCP area and the September-October sea

ice in the west of the Beaufort Sea, with correlation coefficient = 0.51, was revealed. The heavy September-October sea ice

over the west of the Beaufort Sea strongly intensified the early winter haze pollution over the NCP area, or more precisely,

increased the number of haze days. Associated physical mechanisms were further examined. Due to the high albedo and

efficient reflections, the local SST in October became cooler than the climate mean state, showing the radiative cooling



effect. The responses of the atmospheric circulations initially manifested as anomalous cyclonic circulation in the upper

troposphere, and then developed into cyclonic and anti-cyclonic circulations on both sides of the Beaufort Sea in the

subsequent November. The decreased surface wind through the Bering Strait could not transport cold seawater to the Bering

Sea as usual and led to a warmer sea surface over the Bering Sea. The reduction of surface wind speed over the Gulf of

Alaska weakened the seawater evaporation and the latent heat release, which conserved more thermal energy in the sea

surface, i.e., positive SST anomalies.

The November SST anomalies over the Bering Sea and Gulf of Alaska acted as a bridge in the close relationship

between the BSISO (R=0.43, exceeding the 99% confidence level) and HDJ$_{NCP}$ (R=0.61). The warmer sea surface efficiently

heated the air above and resulted in significant responses in the atmosphere. In the upper-troposphere, zonal west wind

anomalies prevailed in the mid-high latitudes, and the meridionality of the atmosphere was reduced, indicating the decrease

of the southward cold air activities. A "+–+" Rossby wave-like train propagated from North China and the Japan Sea,

through the Bering Sea and Gulf of Alaska, to the Cordillera Mts. Near the surface, the NPO-like pattern and the negative

SLP anomalies over Eurasia induced southerly anomalies over the coastal area of China, providing a calm and moist

environment for haze formation. In addition, the intensity of the thermal inversion over the NCP area was significantly

enhanced, and the clear, dry and cold air was difficult to transport to the surface. The horizontal and vertical dispersions were

both limited, so the fine particles were apt to accumulate and cause haze pollution.

In this study, the response of the early winter haze days in the North China plain to the autumn Beaufort Sea ice and the

associated physical mechanisms were investigated. As shown in Figure 2, the HDJ$_{NCP}$ was 42.7 days and reached its

maximum in 2015. Thus, the measurements in 2015 were composited after removing the linear trend to verify the results

from the observational analyses (Figure 16). In September-October 2015, there were positive sea ice anomalies on the west

of the Beaufort Sea (Figure 16a), which satisfied the close relationship revealed in this study. Meanwhile, an obviously

warmer SST in November was observed in most of the BA region (Figure 16b), transferring the impacts of the BSISO. As a

result, a weaker East Asia jet stream, an anomalous southerly (Figure 16c), limited horizontal and vertical dispersion

conditions, and moist air (Figure 16d) enhanced the early-winter haze pollution in 2015. However, some questions remain

unanswered and should be investigated with numerical models in future work. For example, the internal dynamic and

thermal processes and how the heavy sea ice (radiative cooling) affected the atmospheric circulations, are not fully

understood. During this work, linear correlation analyses were the main research technique, and the linear relationship was

discovered. In fact, the dynamic–thermodynamic processes in the air-ice interaction are neither straightforward nor

necessarily linear (Zhang et al., 2000; Gao et al., 2015). Considering the contradiction among the results by a single

numerical model (Gao et al., 2015), a multi-model ensemble was required to solve the internal physical mechanisms.

Furthermore, the November SST anomalies over the Bering Sea and Gulf of Alaska were treated as a bridge to connect the



sea ice and the haze pollution. It is necessary to examine whether this bridge was constructed all the time. Particularly, after

2010, haze pollution became more serious. The driven role of the BSISO and the bridge of $SST_{BA}$ needs to be verified.

Moreover, as revealed by the EOF decomposition, the number of haze days in southern China varied differently. Its

relationship with the sea ice in the Arctic is still unclear and needs to be addressed. The significant relationship revealed in

this study and associated previous work potentially improved the monthly prediction of haze pollution. Valuable haze

predictions are urgently needed by the scientific decision-making departments to control haze pollution in China (Wang,

2018).

**Acknowledgements**

This research was supported by the National Natural Science Foundation of China (41705058 and 91744311), the National

Key Research and Development Plan (2016YFA0600703), the CAS–PKU Partnership Program, and the funding of the

Jiangsu Innovation & Entrepreneurship team.

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

**Table and Figures captions**

**Table 1.** The correlation coefficients (CC) between the BSISO (HDJ$_{NCP}$) and SST indices in October, November, and
December. The linear trend was removed. '*' indicates that the CC exceed the 95% confidence level, and '**' indicates that the CC exceed the 99% confidence level. The meanings of the abbreviations were also explained.

**Figure 1.** The spatial pattern (shading) of the first EOF mode (variation contribution: 33%) for HDJ from 1979 to 2015. The black crosses and dots represent the locations of the observation stations. The cross (dot) indicated that the HDJ accounted for more (less) than 70% of the total winter haze days.

**Figure 2.** The variation of (a) HDJ$_{NCP}$ from 1979 to 2015 and (b) daily maximum PM$_{2.5}$ from December to January in 2015 over the NCP area. The error bars in panel (a) represent one standard error among the measured sites.

**Figure 3.** The correlation coefficients (CC) between the HDJ$_{NCP}$ and September-October sea ice concentration from 1979 to 2015, after detrending. The black dots indicate CC exceeding the 95% confidence level (t test). The black box represents the selected Beaufort Sea.



**Figure 4.** Distributions of the September-October sea ice concentration after removal of the linear trend in typical years (i.e.,
the year when |BSISO|>0.8×its standard deviation). The "+" and "–" represent the mathematical sign of the BSISO and
$HDJ_{NCP}$ indices. The ".sig" indicates that the absolute value of the index anomaly was larger than 0.8×its standard deviation.

**Figure 5.** The CC between the BSISO and SST in (a) October, (b) November, and (c) December, from 1979 to 2015, after
detrending. The black dots indicate CCs exceeding the 95% confidence level (t test). The black boxes (WB: west of Beaufort
Sea, BS: Bering Sea and GA: Gulf of Alaska) are the significantly correlated areas, which were used to calculate the SST
indices.

**Figure 6.** The CC between BSISO and September-October (a) geopotential height (shading), wind (arrow) at 500 hPa, (c)
SLP (shade), and surface wind (arrow); and November (b) geopotential height (shading), wind (arrow) at 500 hPa, (d) SLP
(shade), and surface wind (arrow) from 1979 to 2015, after detrending. The white dots indicate CCs exceeding the 90%
confidence level (t test). The black box in (a-d) represents the location of the Beaufort Sea.

**Figure 7.** The distribution of the climate mean surface wind (arrow) in November and the CC between the BSISO and
surface wind speed in November from 1979 to 2015, after detrending. The black dots indicate CC exceeding the 95%
confidence level (t test). The black boxes (RS1 and RS2) are the significantly correlated areas, which were used to calculate
the $WSPD_{RS1}$ and $WSPD_{RS2}$ index.

**Figure 8.** The CC between $WSPD_{RS2}$ and SST in November from 1979 to 2015. The black dots indicate that the CC
exceeded the 95% confidence level (t test). The black box represents the BA area. The linear trend was removed.

**Figure 9.** The CC between $WSPD_{RS1}$ and (a) SST and (b) latent heat flux in November from 1979 to 2015. The black dots
indicate that the CC exceeded the 95% confidence level (t test). The black box represents the BA area. The linear trend was
removed.

**Figure 10.** The variation of the normalized BSISO (blue) and November $SST_{BA}$ (green) from 1979 to 2015, after detrending.

**Figure 11.** The CC between November $SST_{BA}$ and (a) omega at 200 hPa, (b) at 1000 hPa in December and January from
1979 to 2015. The black dots indicate that the CC exceeded the 95% confidence level (t test). The linear trend was removed.
The black box represents the BA area.

**Figure 12.** The CC between the November $SST_{BA}$ and (a) zonal wind at 200 hPa, (b) wind (arrow), geopotential height
(shading) at 500 hPa, (c) wind (arrow), geopotential height (shading) at 850 hPa, (d) surface wind (arrow), and SLP (shading)
in December-January from 1979 to 2015. The black dots indicate that the CC exceeded the 95% confidence level (t test). The
linear trend was removed. The black boxes in panel (b) represent the three anomalous centers at 500 hPa, and the gray and
black boxes in panel (d) represent the negative and positive anomalous centers.

**Figure 13.** The variation of the November normalized $SST_{BA}$ (gray, bar) and area-averaged geopotential height at 500 hPa of
the three anomalous centers (west: black, middle: green, east: orange) from 1979 to 2015, after detrending.

**Figure 14.** The CC between the November $SST_{BA}$ and (a) surface wind (arrow), specific humidity (shading) at 1000 hPa (b)
BLH (shading), thermal inversion potential (contour, solid (dashed) green lines indicate that the positive (negative)
correlations exceeded the 90% confidence level (t test)) from 1979 to 2015. The black dots indicate that the CC exceeded the
90% confidence level (t test). The linear trend was removed. The black boxes represent the NCP area. The thermal inversion
potential was defined as the air temperature at 850 hPa minus SAT.

**Figure 15.** The cross-section (114°–120°E mean) CC between (a) the $HDJ_{NCP}$, (b) November $SST_{BA}$ and omega (shading),
wind (arrow) in December-January from 1979 to 2015. The black dots indicate that the CCs exceeded the 95% confidence
level (t test). The linear trend was removed.

**Figure 16.** The distributions of (a) September-October sea ice concentration in 2015, (b) sea surface temperature in
November 2015, (c) geopotential height (shading) at 500 hPa, wind (arrow) at 200 hPa in December-January 2015, (d)



specific humidity (shading) at 1000 hPa, BLH (black dots indicate that its value is negative), WSPD (contour, solid black lines indicate a negative value) in December-January 2015. The black box in panel (a) represents the location of the Beaufort Sea, and in panel (b) it represents the BA area. The linear trend was removed.

**Table 1.** The correlation coefficients (CC) between the BSISO (HDJ$_{NCP}$) and SST indices in October, November, and December. The linear trend was removed. '*' indicates that the CC exceed the 95% confidence level, and '**' indicates that the CC exceed the 99% confidence level. The meanings of the abbreviations were also explained.

| CC | | Oct | Nov | Dec |
|---|---|---|---|---|
| **BSISO:** Beaufort Sea Ice in Sep-Oct | **SST$_{WB}$**: SST over the west of the Beaufort Sea | −0.75** | −0.26 | −0.23 |
| | **SST$_{BS}$**: SST over the Bering Sea | 0.27 | 0.41* | 0.45** |
| | **SST$_{GA}$**: SST over the Gulf of Alaska | 0.31 | 0.40* | 0.44** |
| | **SST$_{BA}$**: SST$_{BS+}$ SST$_{GA}$ | 0.34* | 0.43** | 0.48** |
| **HDJ$_{NCP}$:** Haze days in Dec-Jan | **SST$_{WB}$**: SST over the west of the Beaufort Sea | −0.30 | −0.05 | −0.10 |
| | **SST$_{BA}$:** SST$_{BS+}$ SST$_{GA}$ | 0.52** | 0.61** | 0.56** |

**Figure 1. The spatial pattern (shading) of the first EOF mode (variation contribution: 33%) for HDJ from 1979 to 2015. The black**

**crosses and dots represent the locations of the observation stations. The cross (dot) indicated that the HDJ accounted for more (less)**

**than 70% of the total winter haze days.**



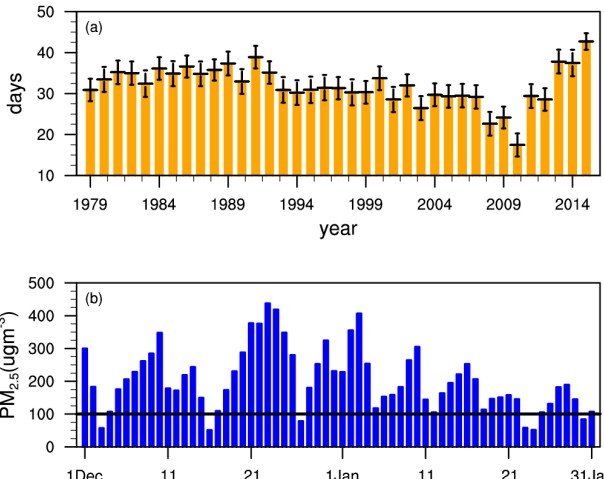

**Figure 2. The variation of (a) HDJNCP from 1979 to 2015 and (b) daily maximum PM2.5 from December to January in 2015 over the NCP area. The error bars in panel (a) represent one standard error among the measured sites.**

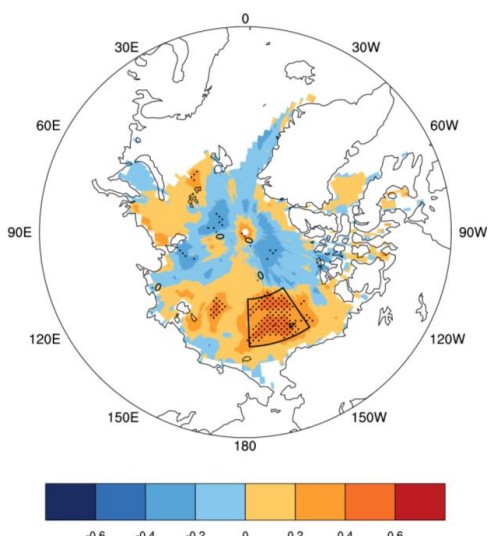


**Figure 3. The correlation coefficients (CC) between the HDJNCP and September-October sea ice concentration from 1979 to 2015, after detrending. The black dots indicate CC exceeding the 95% confidence level (t test). The black box represents the selected Beaufort Sea.**





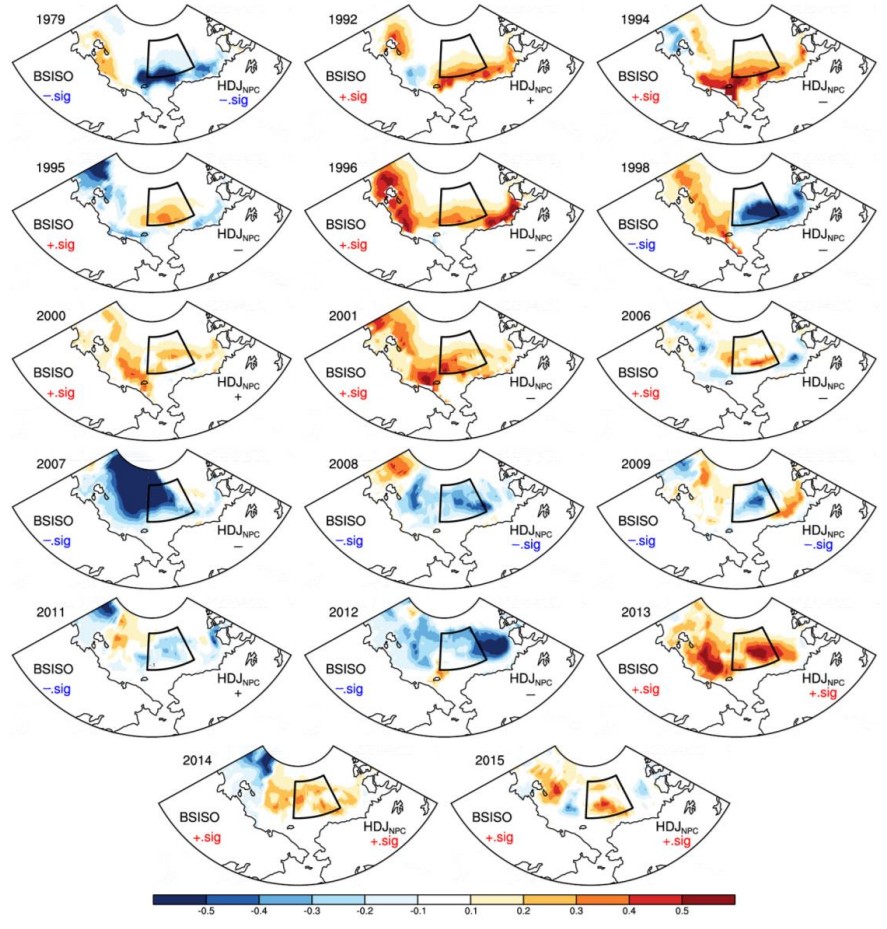

**Figure 4. Distributions of the September-October sea ice concentration after removal of the linear trend in typical years (i.e., the year when |BSISO|>0.8×its standard deviation). The "+" and "−" represent the mathematical sign of the BSISO and HDJNCP indices. The ".sig" indicates that the absolute value of the index anomaly was larger than 0.8×its standard deviation.**





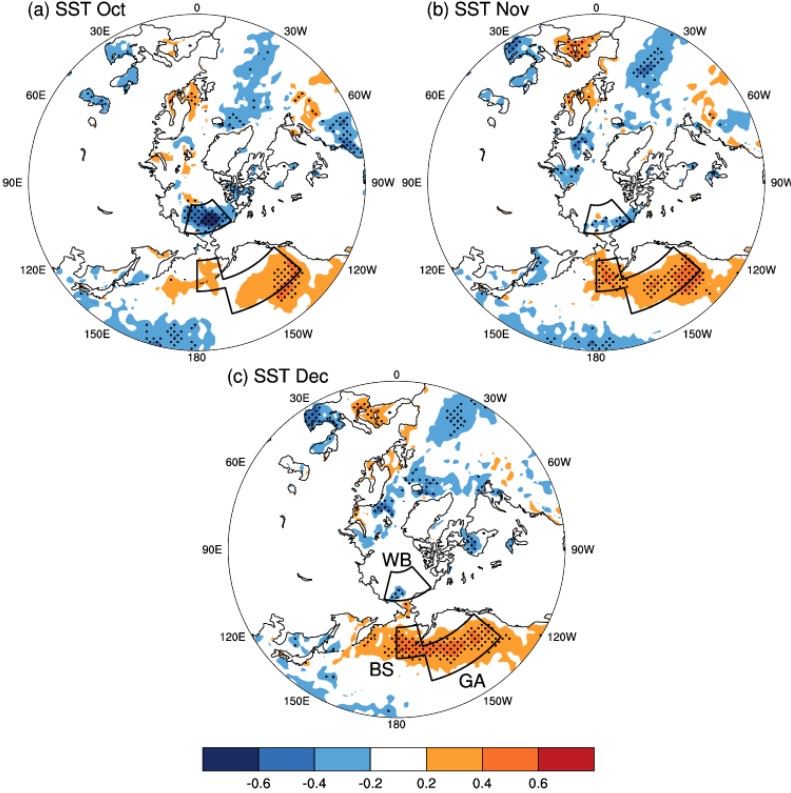

**Figure 5. The CC between the BSISO and SST in (a) October, (b) November, and (c) December, from 1979 to 2015, after detrending.**

**The black dots indicate CCs exceeding the 95% confidence level (t test). The black boxes (WB: west of Beaufort Sea, BS: Bering Sea and GA: Gulf of Alaska) are the significantly correlated areas, which were used to calculate the SST indices.**

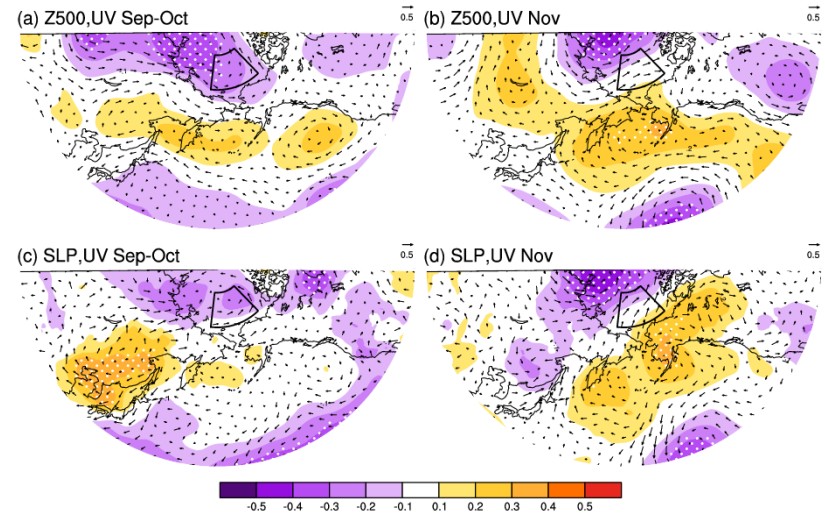




**Figure 6. The CC between BSISO and September-October (a) geopotential height (shading), wind (arrow) at 500 hPa, (c) SLP (shade), and surface wind (arrow); and November (b) geopotential height (shading), wind (arrow) at 500 hPa, (d) SLP (shade), and**

**surface wind (arrow) from 1979 to 2015, after detrending. The white dots indicate CCs exceeding the 90% confidence level (t test). The black box in (a-d) represents the location of the Beaufort Sea.**

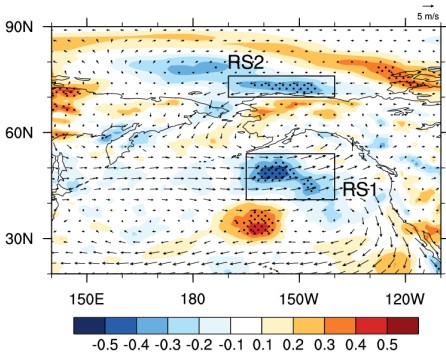

**Figure 7. The distribution of the climate mean surface wind (arrow) in November and the CC between the BSISO and surface wind speed in November from 1979 to 2015, after detrending. The black dots indicate CC exceeding the 95% confidence level (t test). The**

**black boxes (RS1 and RS2) are the significantly correlated areas, which were used to calculate the WSPDRS1 and WSPDRS2 index.**

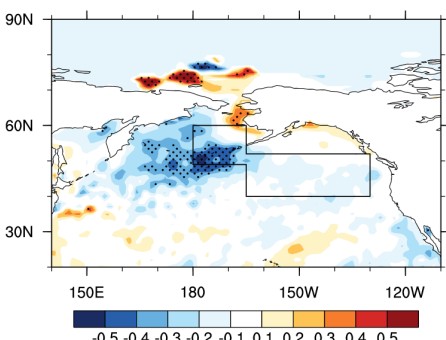

**Figure 8. The CC between WSPDRS2 and SST in November from 1979 to 2015. The black dots indicate that the CC exceeded the 95% confidence level (t test). The black box represents the BA area. The linear trend was removed.**





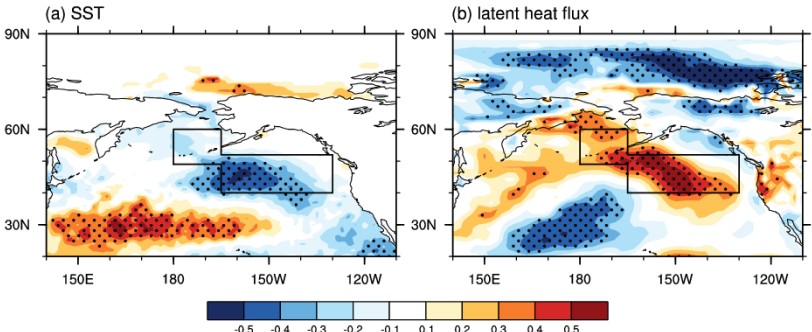

**Figure 9. The CC between WSPDRS1 and (a) SST and (b) latent heat flux in November from 1979 to 2015. The black dots indicate that the CC exceeded the 95% confidence level (t test). The black box represents the BA area. The linear trend was removed.**

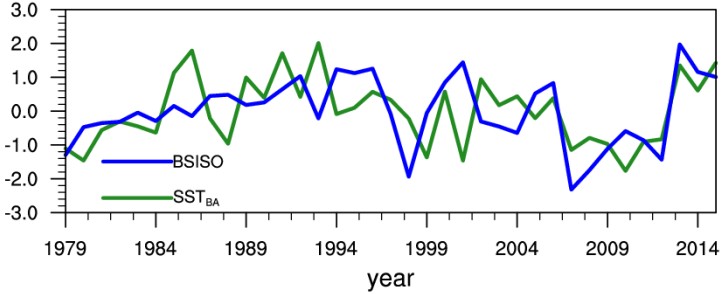

**Figure 10. The variation of the normalized BSISO (blue) and November SSTBA (green) from 1979 to 2015, after detrending.**

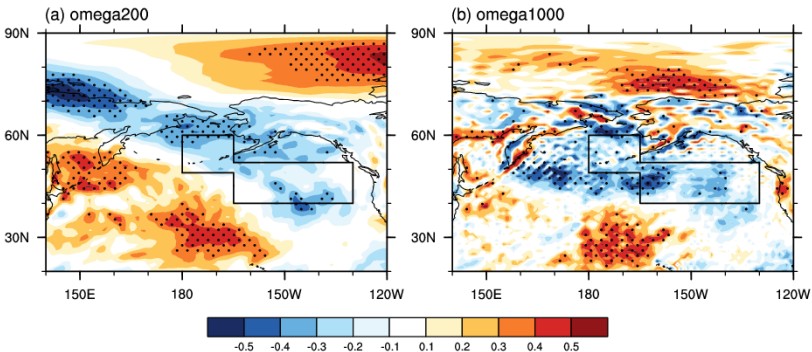

**Figure 11. The CC between November SSTBA and (a) omega at 200 hPa, (b) at 1000 hPa in December and January from 1979 to 2015. The black dots indicate that the CC exceeded the 95% confidence level (t test). The linear trend was removed. The black box represents the BA area.**





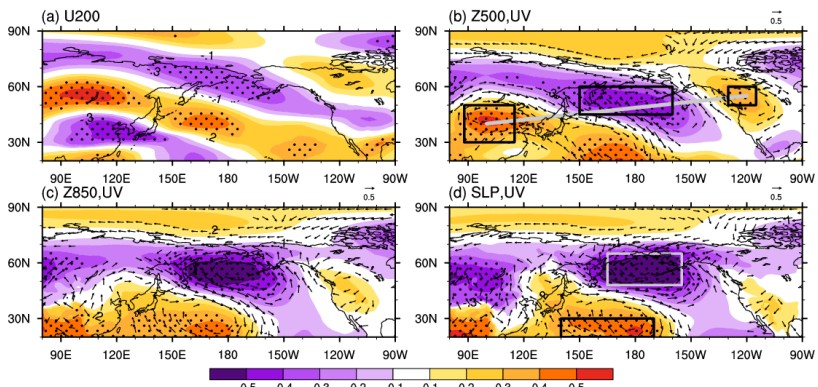

**Figure 12. The CC between the November SSTBA and (a) zonal wind at 200 hPa, (b) wind (arrow), geopotential height (shading) at**

**500 hPa, (c) wind (arrow), geopotential height (shading) at 850 hPa, (d) surface wind (arrow), and SLP (shading) in**

**December-January from 1979 to 2015. The black dots indicate that the CC exceeded the 95% confidence level (t test). The linear**

**trend was removed. The black boxes in panel (b) represent the three anomalous centers at 500 hPa, and the gray and black boxes in**

**panel (d) represent the negative and positive anomalous centers.**

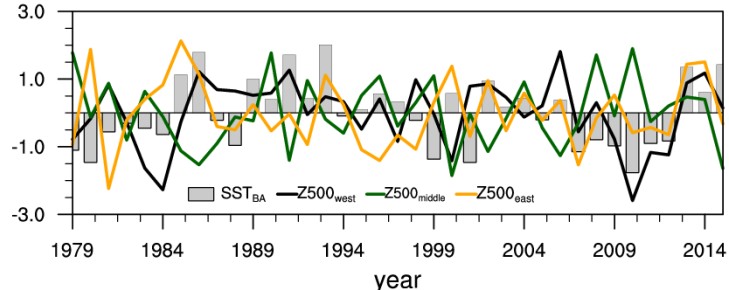

**Figure 13. The variation of the November normalized SSTBA (gray, bar) and area-averaged geopotential height at 500 hPa of the**

**three anomalous centers (west: black, middle: green, east: orange) from 1979 to 2015, after detrending.**





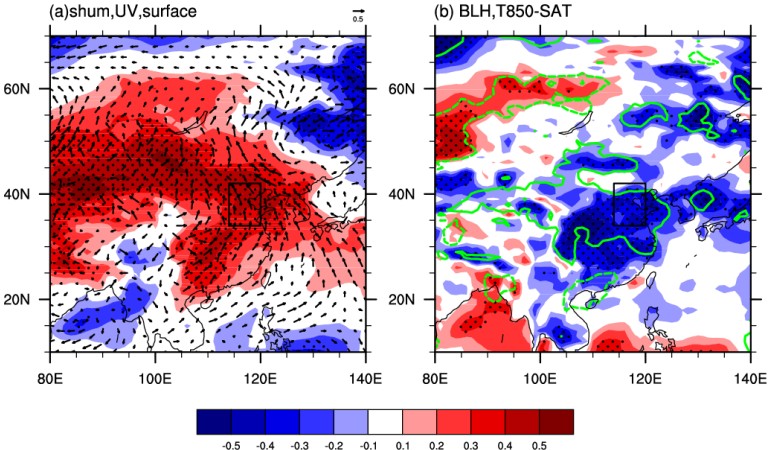

**Figure 14. The CC between the November SSTBA and (a) surface wind (arrow), specific humidity (shading) at 1000 hPa (b) BLH**

**(shading), thermal inversion potential (contour, solid (dashed) green lines indicate that the positive (negative) correlations exceeded**

**the 90% confidence level (t test)) from 1979 to 2015. The black dots indicate that the CC exceeded the 90% confidence level (t test).**

**The linear trend was removed. The black boxes represent the NCP area. The thermal inversion potential was defined as the air**

**temperature at 850 hPa minus SAT.**

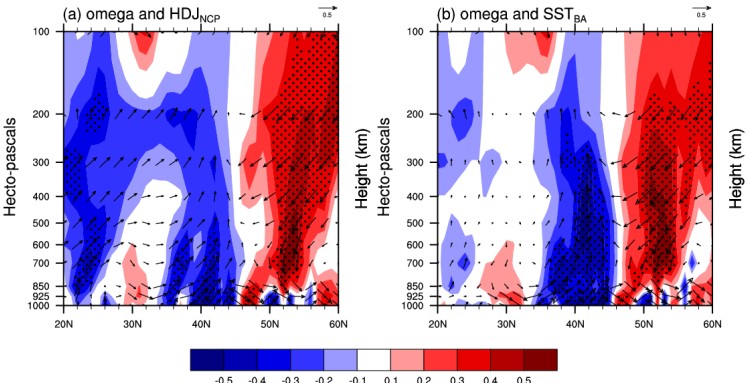

**Figure 15. The cross-section (114 °–120 °E mean) CC between (a) the HDJNCP, (b) November SSTBA and omega (shading), wind**

**(arrow) in December-January from 1979 to 2015. The black dots indicate that the CCs exceeded the 95% confidence level (t test).**

**The linear trend was removed.**



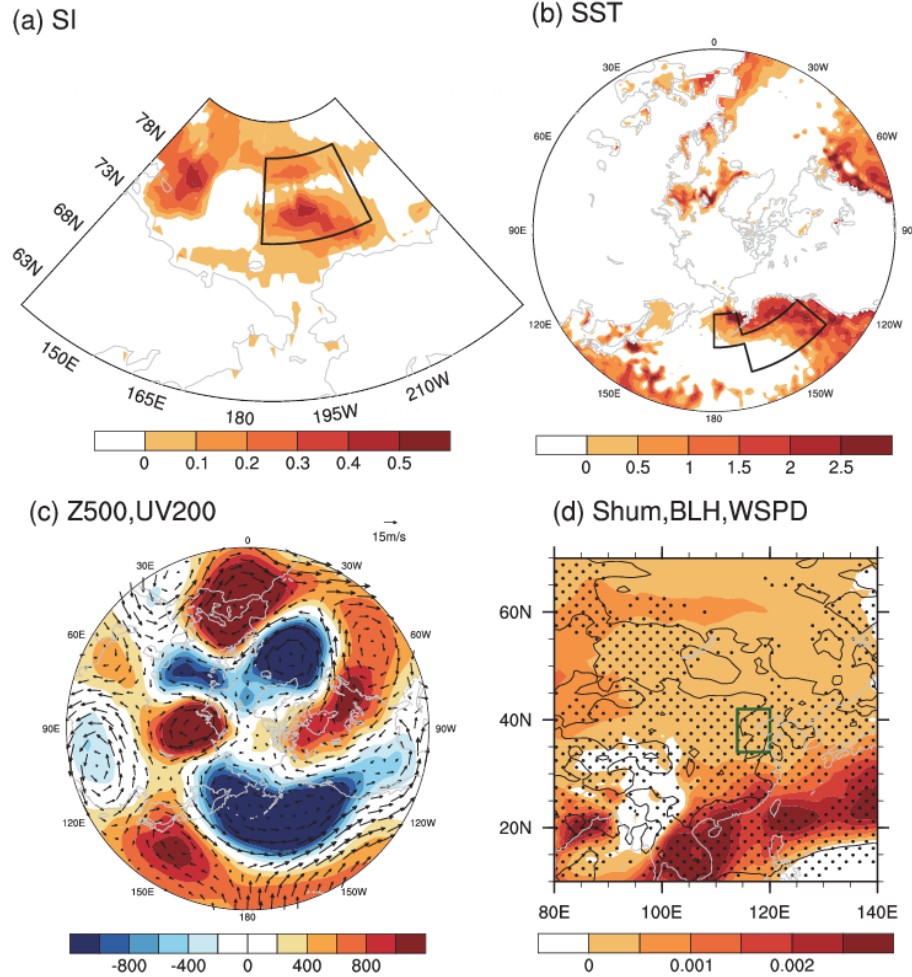

**Figure 16. The distributions of (a) September-October sea ice concentration in 2015, (b) sea surface temperature in November 2015, (c) geopotential height (shading) at 500 hPa, wind (arrow) at 200 hPa in December-January 2015, (d) specific humidity (shading) at 1000 hPa, BLH (black dots indicate that its value is negative), WSPD (contour, solid black lines indicate a negative value) in December-January 2015. The black box in panel (a) represents the location of the Beaufort Sea, and in panel (b) it represents the BA area. The linear trend was removed.**
