# Peer review of "Response of Early Winter Haze Days in the North China Plain to Autumn Beaufort Sea Ice"

_Atmospheric Chemistry and Physics, 2018_

## Referee Comment (RC1) · Anonymous Referee #1 · 20 Nov 2018

Review of "Response of early winter haze days in the North China plain to autumn Beaufort sea ice" by Yin et al. (MS: ACP-2018-783)

Summary: Yin et al. have found a high correlation (0.51) between the early winter haze days in the North China plain and the September-October sea ice in the west of the Beaufort Sea. Further analysis revealed that the sea surface temperature anomalies over the Bering Sea and Gulf of Alaska acting as a bridge that linked the variations of haze days and sea ice. This is interesting, also important for us to understand the causing of the changes of haze pollutions over China in recent years. I recommend it to be accepted by ACP after several corrections.

1. In recent years, there are increasing works referring to the impact of climate change on the haze pollution over China. The authors should present updating review on

these new papers in the introduction. 2. Line 46: The reference here is not found in the reference list. "2017" may be "2016"? 3. Some information for the site observation should be clear. For example, how many meteorological sites used here? as well as the number of monitoring sites for PM2.5. How to deal with the missing values. 4. The definition of the haze pollution should be clear. 5. Line 84-85: This expression here is not correct. Here, just the number of haze days is highlighted, not the synoptic process. 6. Line 90-92: The linear trend here has been deleted or not? It should be clear here as well as in the figure caption. 7. Line 105: the "heavy" used here is not correct, as well as in the other places throughout MS. 8. As we all know, the wind is one of key factor that exerts impact on the haze pollution. Compared to the zonal wind, the meridional wind generally performs a greater role on the particulate dissipation. So, the influence of the sea ice on the meridional wind should be checked. 9. What about the relationship between the local wind speed and Beaufort Sea ice/SST anomalies over the Bering Sea and Gulf of Alaska? 10. The English writing should be further improved.

––––––––––––––––––––––––––––––

---

## Referee Comment (RC2) · Anonymous Referee #2 · 30 Nov 2018

General comments: Haze pollution is for the time being a serious problem for China. The prediction of haze pollution is highly-relevant to the society. The manuscript explored the linkage between the number of haze days in China and the change in autumn sea ice extent in the Beaufort Sea and analyze the potential mechanism. I find the manuscript is scientifically interesting and fits the scope of ACP. However, major revision is needed before it can be accepted for publication in ACP.

Major comment: 1. the cause and effect are not convincing in this manuscript. There are quite some places authors used 'induced'. Correlation/regression can not tell what is cause and what is the effect. 2. Why can not directly link the SST anomalies in the Bering Sea and the number of haze days over NCP? Detailed comments: 1. What is the difference between 'Arctic region' (Line 23) and 'Arctic area' (Line 25)? 2. Line 23, I

do not understand why authors highlighted 'February 2018' since no data from 2018 is used in the manuscript. 3. Line 24, Does the authors mean the Arctic amplification intensified only during past few years? 4. Line 26, What the authors mean by 'Recently'? 5. Line 26, 'Arctic sea ice decreases rapidly since the satellite era, in particular, after year 2000'. 6. Line 27, 'the change of ASI' 7. Line 30, remove 'variability' 8. Line 30, sea ice is a component in climate system, is not an external driver 9. Line 40, it is better if the authors can provide a brief definitions for dust, sandstorm and haze. 10. Line 45, 'long-term trend of haze' is not clear. Long-term trend of number of haze days, or intensity of haze, or periods of haze? 11. Line 46, the same as above 'the trend of haze pollution' 12. Line 50, the same as above, 'correlation with the haze' 13. Line 51, the same as above, 'different variations in haze days' 14. Line 52, 'between the autumn sea ice cover in Beaufort Sea and the number of haze days in winter' 15. Line 54. 'number of haze days varied differently during early (December-January) and late (February) winters' 16. Line 54', 'suggesting a potential different driving mechanism' 17. Line 56, similar as No. 14 18. Line 80', 'The HDJ was stable during 1979 to 2012 and decreased during 1993 to 2009' 19. Line 80, 'The HDJ showed a strong upward trend after 2009' 20. Line 85, 'what is the threshold for pollution in China'? 21. Line 84, what is the meaning of 'synoptic process of haze were weaker'? How to judge this? 22. Line 90, there are number of places the authors used haze days. I believe that authors mean 'number of haze days'. 23. Line 97, the correlation can not tell which causes which. 24. Line 101, 'correspond' instead of 'induce'. Again, correlation can not tell which causes which 25. Line 102, can authors perform sliding correlation to indicate the enhanced connection after mid-1990? 26. Line 103, 'response' is not accurate here 27. Line 108, SST and sea ice concentration in general co-varies. Correlation can not tell which causes which. Authors can also check the surface heat flux. 28. Line 108, 'induced' is not correct here. 29. Line 109, Do authors have any idea why negative SST anomalies disappear in November? 30. Line 110, correlation can not tell 'change of BS sea ice' can lead to SST anomalies over the BS and GA. 'induced' is not correct here. 31. Line 118, why authors can not directly link the SST anomalies over BS and

GA to HDJ? 32. Line 123, how authors can conclude the change in atmosphere circulation is a response to change in sea ice by correlation? 33. Line 125, 'induced' again 34. After line 125, authrs used correlation to conclude the sea ice change-leading to atmosphere change-leading to SST change in number of places of the manuscript.

---

## Author Comment (AC1) · 5 Jan 2019

**Response to Review #1**

Summary: Yin et al. have found a high correlation (0.51) between the early winter haze days in the North China plain and the September-October sea ice in the west of the Beaufort Sea. Further analysis revealed that the sea surface temperature anomalies over the Bering Sea and Gulf of Alaska acting as a bridge that linked the variations of haze days and sea ice. This is interesting, also important for us to understand the causing of the changes of haze pollutions over China in recent years. I recommend it to be accepted by ACP after several corrections.

1. **In recent years, there are increasing works referring to the impact of climate change on the haze pollution over China. The authors should present updating review on these new papers in the introduction.**

*Reply:*

The impact of climate change on haze pollution in China was a meaningful scientific issue and were paid attentions in recent years. Some related publications were cited now to update the introduction.

*Revision:*

…For the long-term trend of number of haze days, human activities are the recognized and fundamental driver (Li et al., 2018; Yang et al 2016; Chen et al., 2018; Zhang et al., 2018)…

…By the sensitive experiments, Li et al. (2017) emphasized the impacts of ASI anomalies on haze pollution in North China, but deemphasized the role of ENSO (He et al., 2019)…

Supplemented new papers:

Chen, H. P., Wang, H. J., Sun, J. Q., Xu, Y. Y., and Yin, Z. C.: Anthropogenic Fine Particulate Matter Pollution Will Be Exacerbated in Eastern China Due to 21st-Century GHG Warming, Atmos. Chem. Phys. Discuss., https://doi.org/10.5194/acp-2018-761, in review, 2018.

He, C., Liu, R., Wang, X. M., Liu, S. C., Zhou, T. J., Liao, W. H.: How does El

Niño-Southern Oscillation modulate the interannual variability of winter haze days over eastern China? Science of the Total Environment, 651, 1892–1902, https://doi.org/10.1016/j.scitotenv.2018.

10.100, 2019.

Zhang, Q. Q., Ma, Q., Zhao, B., et al.: Winter haze over North China Plain from 2009 to 2016: Influence of emission and meteorology, Environmental Pollution, 242: 1308–1318, 2018.

**2. Line 46: The reference here is not found in the reference list. "2017" may be "2016"?**

*Reply:*

The error was revised.

*Revision:*

…but the rapid ASI decline also contributed to the trend of number of haze days in the North China Plain after 2000 (Wang and Chen 2016)…

**3. Some information for the site observation should be clear. For example, how many meteorological sites used here? as well as the number of monitoring sites for PM2.5. How to deal with the missing values.**

*Reply:*

(1) The sites used here was shown in Figure 1. The cross (dot) indicated that the HDJ accounted for more (less) than 70% of the total winter haze days.

[Figure]

(2) More information were added. The number of meteorological sites were used to calculated the $HDJ_{NCP}$ was 38, and the number of $PM_{2.5}$ sites were 162.

(3) The sites with missing values >5% was discarded, the others were kept in the datasets.

*Revision:*

In this study, we focused on the HDJ in the NCP region ($HDJ_{NCP}$, i.e., mean of the 38 sited HDJ) and its connection with the autumn ASI.

The hourly $PM_{2.5}$ concentration data were provided by the Ministry of Environmental Protection of China, including 162 sites in the North China.

**4. The definition of the haze pollution should be clear.**

*Reply:*

The definition of haze was added in the manuscripts.

*Revision:*

That is, if the visibility was lower than 10km and the relative humidity was drier than 90%, the day was defined as one haze day after filtering the other weather affected visibility (i.e., precipitation, dust, sandstorm, etc.).

**5. Line 84-85: This expression here is not correct. Here, just the number of haze days is highlighted, not the synoptic process.**

*Reply:*

The error was revised.

*Revision:*

The daily maximum of area-mean PM$_{2.5}$ in 2015 is shown in Figure 2b and was above 100 μg/m³. The  concentrations of PM$_{2.5}$ were relatively lower in January 2016 than those in December but still exceeded the threshold of pollution in China (i.e., 75 μg/m³). On 23 December, the most disastrous haze occurred, and the area-mean

**6. Line 90-92: The linear trend here has been deleted or not? It should be clear here as well as in the figure caption.**

*Reply:*

To emphasize the interannual variation, the linear trend was removed.

*Revision:*

…the correlation coefficients between the HDJ$_{NCP}$ and the September-October sea ice were assessed after removing the linear trend (Figure 3)…

**7. Line 105: the "heavy" used here is not correct, as well as in the other places throughout MS.**

*Reply:*

The error was revised throughout the MS.

*Revision:*

The  positive sea ice anomalies, with high albedo, can efficiently reflect solar radiation and restore more fresh water, which could influence the local and adjacent SST. The correlation coefficients between BSISO and the simultaneous and subsequent SST were computed (Figure 5). Because of efficient reflections of the solar radiation, the locally negative SST anomalies, located near the west of the Beaufort Sea (70–81°N, 166°E–138°W), were associated with the  positive BSISO anomalies in October. In the following two months, these negative SST anomalies could not be sustained, i.e., these anomalous responses disappeared in November. However, the  positive SST anomalies in the

**8. As we all know, the wind is one of key factor that exerts impact on the haze pollution. Compared to the zonal wind, the meridional wind generally performs a greater role on the particulate dissipation. So, the influence of the sea ice on the meridional wind should be checked.**

*Reply:*

The arrows in Figure 14a was the influences of the $SST_{BA}$ on the surface wind and has included the meridional wind.

Furthermore, we also plotted the required Figures, but did not repeat it in the manuscript. The $SST_{BA}$ was the bridge to connect the number of haze days and the sea ice anomalies. In the flowing Figure, the surface meridional wind showed significantly positive correlation with the $SST_{BA}$. The positive correlation indicate enhanced southerly anomalies, which weakened the cold air from the high latitude and make the dissipation conditions poor. Thus, the haze occurred easily.

[Figure]

Figure. The CC between the November $SST_{BA}$ and surface meridional wind, the black dots indicate that the CC exceeded the 95% confidence level

*Revision:*

cold air and ventilation conditions. Compared to the local zonal wind, the meridional wind played more important roles on weakening the horizontal dissipation conditions of the air (Figure omitted). The  southerly anomalies were located over the coastal area of China and transported moisture to the NCP area (Figure 14a), providing moist air for haze formation.

**9. What about the relationship between the local wind speed and Beaufort Sea ice/SST anomalies over the Bering Sea and Gulf of Alaska?**

*Reply:*

We plotted the required Figures as follows. It is obvious that the relationship between the local wind speed and Beaufort Sea ice/SST anomalies over the Bering Sea and Gulf of Alaska was negative, but the relationship was not as significant as the meridional wind. (1) The wind speed anomalies associated with preceding sea ice/SST anomalies were negative. The smaller wind speed indicate poor horizontal dissipation conditions in the air, thus the particulates accumulated efficiently. (2) In weakening the horizontal dissipation, the changed of the meridional wind played more important roles.

[Figure]

Figure. The CC between the (a) BSISO, (b) November $SST_{BA}$ and surface meridional wind, the black dots indicate that the CC exceeded the 95% confidence level

*Revision:*

cold air and ventilation conditions. Compared to the local zonal wind, the meridional wind played more important roles on weakening the horizontal dissipation conditions of the air (Figure omitted). The induced southerly anomalies were located over the coastal area of China and transported moisture to the NCP area (Figure 14a), providing moist air for haze formation.

**10. The English writing should be further improved.**

*Reply:*

The English has been improved the native speaker.

---

## Author Comment (AC2) · 5 Jan 2019

**Response to Review #2**

General comments: Haze pollution is for the time being a serious problem for China. The prediction of haze pollution is highly-relevant to the society. The manuscript explored the linkage between the number of haze days in China and the change in autumn sea ice extent in the Beaufort Sea and analyze the potential mechanism. I find the manuscript is scientifically interesting and fits the scope of ACP. However, major revision is needed before it can be accepted for publication in ACP

Major comment:

**1. the cause and effect are not convincing in this manuscript. There are quite some places authors used 'induced'. Correlation/regression can not tell what is cause and what is the effect.**

*Reply:*

(1) To verify the proposed causality, numerical experiments were designed by the public CESM-LE datasets. **A new section "5. Causality verification by CESM-LE experiments" and a new Figure 16 was added in the manuscript.**

[revised manuscript text omitted]

**2. Why can not directly link the SST anomalies in the Bering Sea and the number of haze days over NCP?**

*Reply:*

Certainly, there was directly link between the November SST anomalies and the number of haze days in December and January. There are two reasons why we link the number of haze days and the September-October sea ice. (1) As an efficient driver, the September-October sea ice was one month in advance of the November SST, which **supports sufficient time gap to make the seasonal prediction in the real-time operation.** That is, in November, we may gain the sea ice September-October data and run the statistical seasonal prediction models. (2) The goal of this manuscript is to reveal the connection between the sea ice and the haze pollution in the early winter. Our studies **not only reveal the link between the SST and haze, but also deepen the understanding the impacts of the sea ice on the haze** by taking the SST anomalies as a bridge.

Detailed comments:

**1. What is the difference between 'Arctic region' (Line 23) and 'Arctic area' (Line 25)?**

*Reply:*

The presentations were coalesced to "Arctic region".

*Revision:*

 During the past few years, the increase of surface air temperature has been distinctly amplified in the Arctic  region(i.e., the Arctic Amplification feature) and approximately

**2. Line 23, I do not understand why authors highlighted 'February 2018' since no data from 2018 is used in the manuscript.**

*Reply:*

In the old version, we mentioned 'February 2018' to emphasize the importance of Arctic sea ice. Now, to focus on the scientific issue, the associated texts were deleted.

*Revision:*

In February 2018, the highest temperature near the Arctic region was above the freezing point (Jason, 2018), raising tremendous concerns from global climate change scientists. During the past few years, the increase of surface air temperature has been distinctly amplified in the Arctic area region(i.e., the Arctic Amplification feature) and approximately

**3. Line 24, Does the authors mean the Arctic amplification intensified only during past few years?**

*Reply:*

Our presentation was confusing. We did not mean the Arctic amplification intensified. We wanted to introduce that the increase of surface air temperature has been distinctly amplified in the Arctic region and lead to the definition of Arctic amplification. The confusing texts have been revised.

*Revision:*

tremendous concerns from global climate change scientists. During the past few years, the increase of surface air temperature has been distinctly amplified in the Arctic area region(i.e., the Arctic Amplification feature) and approximately twice as large as the average increase in global warming, which was called the Arctic Amplification (Zhou, 2017). Recently,

**4. Line 26, What the authors mean by 'Recently'?**

**5. Line 26, 'Arctic sea ice decreases rapidly since the satellite era, in particular, after year 2000'.**

*Reply:*

According to the advice of the reviewer, detailed comments 4 &5 were revised together.

*Revision:*

…Arctic sea ice (ASI) decreases rapidly since the satellite era, in particular, after the year of 2000 (Gao et al., 2015)…

twice as large as the average increase in global warming, which was called the Arctic Amplification (Zhou, 2017). Recently, the Arctic sea ice (ASI) decreased rapidly due to the Arctic Amplification and reached a record low in September 2012since the satellite era, in particular, after the year of 2000 (Gao et al., 2015). The loss change of ASI, associated with changed the

**6. Line 27, 'the change of ASI'**

*Reply:*

According to the advice of the reviewer, the errors were revised.

*Revision:*

…The change of ASI, associated with changed reflection of solar radiation and the exchange of energy and fresh water, could remotely connect with the climate in the Northern Hemisphere, especially the winter climate in Eurasia…

**7. Line 30, remove 'variability'**

*Reply:*

According to the advice of the reviewer, the errors were revised.

*Revision:*

…especially the winter climate in Eurasia…

**8. Line30, sea ice is a component in climate system, is not an external driver**

*Reply:*

According to the advice of the reviewer, the errors were revised.

*Revision:*

climate in the Northern Hemisphere, especially the winter climate  in Eurasia (Liu et al., 2007; Wang and Liu, 2016). The decreased ASI over the Barents–Kara Seas in late autumn stimulated a planetary-scale Rossby wave train in early winter (Honda et al., 2009; Kim et al., 2014) and transported its

**9. Line 40, it is better if the authors can provide a brief definitions for dust, sandstorm and haze.**

*Reply:*

The brief definitions for dust, sandstorm and haze were provided.

*Revision:*

…The dust (dry particles suspended in air after strong wind) and sandstorm (strong wind carrying sand) over North China, types of weather that are sensitive to wind, also showed close relationships with the variation of ASI after the mid-1990s (Fan et al., 2017)…

…Haze (polluted particulate aerosols suspended in air), also being sensitive to wind, frequently occurred under calm and static weather conditions…

**10. Line 45, 'long-term trend of haze' is not clear. Long-term trend of number of haze days, or intensity of haze, or periods of haze?**

**11. Line 46, the same as above 'the trend of haze pollution'**

**12. Line 50, the same as above, 'correlation with the haze'**

**13. Line51, the same as above, 'different variations in haze days'**

**14. Line 52, 'between the autumn sea ice cover in Beaufort Sea and the number of haze days in winter'**

**15. Line54. 'number of haze days varied differently during early (December-January) and late (February) winters'**

*Reply:*

The presentation "haze days" was confusing and should be **the number of the haze days**. Detailed comments 10–15 were revised together.

*Revision:*

Haze (polluted particulate aerosols suspended in air), also being sensitive to wind, frequently occurred under calm and static weather conditions, i.e., small surface winds and strong thermal inversion (Yin et al., 2015; Ding and Liu, 2014; Chen and Wang, 2015; Cai et al., 2017; Gao and Chen, 2017). For the long-term trend of number of haze days, human activities are the recognized and fundamental driver (Li et al., 2018; Yang et al 2016; Chen et al., 2018; Zhang et al., 2018), but the rapid ASI decline also contributed to the trend of number of haze days in the North China Plain after 2000 (Wang and Chen 20167). For the interannual to interdecadal variations, the impacts of ASI on the number of haze days in the east of China were emphasized by observational analyses (Wang et al., 2015) and numerical studies (Li et al. 2017). By the sensitive experiments, Li et al. (2017) emphasized the impacts of ASI anomalies on haze pollution in North China, but deemphasized the role of ENSO (He et al., 2019). From 1979–2012, the ASI loss led to a northward shift of the East Asia jet stream and weak East Asian winter monsoons, indicating a strongly negative correlation with the number of haze days in the east of China (Wang et al. 2015). However, the first mode of the Empirical Orthogonal Function (EOF) in Yin and Wang (2016a) presented different variations of number of haze days in the south and north of the Yangtze River. The positive relationship between the autumn sea ice in the Beaufort Sea and the number of haze days in winter was briefly revealed without sufficient physical explanations but contributed to the prediction of number of haze days in winter (Yin and Wang 2016b, 2017b). The number of haze days in early winter (December-January)  also varied differently with that in February  (figure omitted),

**16. Line 54', 'suggesting a potential different driving mechanism'**

*Reply:*

According to the advice of the reviewer, the errors were revised.

*Revision:*

…The number of haze days in early winter (December-January) also varied differently with that in February (figure omitted), suggesting a potential different driving mechanism…

17. Line 56, similar as No. 14

*Reply:*

The presentation "haze days" was confusing and should be the number of the haze days.

*Revision:*

…Thus, an open question still existed, i.e., the connections between Beaufort Sea ice (BSI) and the number of haze days in early winter in the North China Plain (NCP: 34–42$^{\circ}$N, 114-120$^{\circ}$E) and the associated physical mechanisms…

**18. Line 80', 'The HDJ was stable during 1979 to 2012 and decreased during 1993 to 2009'**

*Reply:*

According to the advice of the reviewer, the confusing presentation were revised.

*Revision:*

…The HDJ$_{NCP}$ was stable during 1979 to 1992 and decreased from 1993 to 2009…

**19. Line 80, 'The HDJ showed a strong upward trend after 2009'**

*Reply:*

According to the advice of the reviewer, the confusing presentation were revised.

*Revision:*

…After 2009, the HDJ$_{NCP}$ showed a strong upward trend…

**20. Line 85, 'what is the threshold for pollution in China'?**

*Reply:*

The threshold was supplemented.

*Revision:*

…still exceeded the threshold of pollution in China (i.e., 75 μg/m³)…

**21. Line 84, what is the meaning of 'synoptic process of haze were weaker'? How to judge this?**

*Reply:*

The "synoptic processes of haze" was not accurate. According to the advice of the reviewer, the confusing presentation were revised.

"synoptic processes of haze"→"the concentrations of PM$_{2.5}$"

*Revision:*

The daily maximum of area-mean PM$_{2.5}$ in 2015 is shown in Figure 2b and was above 100 µg/m³. The   concentrations of PM$_{2.5}$ were relatively lower in January 2016 than those in December but still exceeded the threshold of pollution in China (i.e., 75 µg/m³). On 23 December, the most disastrous haze occurred, and the area-mean

**22. Line 90, there are number of places the authors used haze days. I believe that authors mean 'number of haze days'.**

*Reply:*

The haze days were revised to the "number of haze days" throughout the MS.

*Revision:*

of China by modulating the large-scale atmospheric circulations and local meteorological conditions. Furthermore, the opposite pattern of number of haze days in the east of China was revealed in Figure 1. To confirm the response of

**23. Line 97, the correlation cannot tell which causes which.**

**24. Line 101, 'correspond' instead of 'induce'. Again, correlation cannot tell which causes which**

**26. Line 103, 'response' is not accurate here**

**27. Line 108, SST and sea ice concentration in general co-varies. Correlation can not tell which causes which. Authors can also check the surface heat flux.**

**28. Line 108, 'induced' is not correct here.**

**30. Line 110, correlation can not tell 'change of BS sea ice' can lead to SST anomalies over the BS and GA. 'induced' is not correct here.**

**32. Line 123, how authors can conclude the change in atmosphere circulation is a response to change in sea ice by correlation?**

**33. Line 125, 'induced' again**

**34. After line 125, authrs used correlation to conclude the sea ice change-leading to atmosphere change-leading to SST change in number of places of the manuscript.**

*Reply:*

Detailed comments 23, 24, 26–28, 30, 32–34 concentrated on the meaning of the correlation method and were similar with the Major comment 1.

(1) During the statistical analysis sections, the presentations, like "induce", "response", were modified.

(2) To verify the proposed causality, numerical experiments were designed by the public CESM-LE datasets. **A new section "5. Causality verification by CESM-LE experiments" and a new Figure 16 was added.**

(3) Analysis about the surface heat flux was done in Figure 9. As follows:

[revised manuscript text omitted]

**25. Line 102, can authors perform sliding correlation to indicate the enhanced connection after mid-1990?**

*Reply:*

The sliding (21-yr running) correlation was plotted in Figure R2. It is obvious that the correlation coefficient was insignificant before 2000, but became significant then. The correlation coefficient during 1980–1997 was 0.11, but was 0.55 during 1998–2015. Furthermore, the number of years when the anomalies of $HDJ_{NCP}$ and BSISO with the same mathematical sign ($NY_{SMS}$) were counted and those with significant amplitudes (i.e., |anomalies| > 0.8 × its standard deviation) among the $NY_{SMS}$ values were extracted and termed $NY_{SA}$. Compared to P1, both $NY_{SMS}$ and $NY_{SA}$ significantly increased during P2. Specifically, there were 8 (0) $NY_{SMS}$ ($NY_{SA}$) years before the mid-1990s, which dramatically increased to 13 (5) years during 1998–2015 (Figure R3).

To answer the reasons for the change of the correlationship, a paper titled "Enhanced Contributions of Beaufort Sea Ice to Early-winter Haze Days in North China after the mid-1990s" was prepared.

[Figure]

Figure R2. The 21-yr running correlation coefficient between BSISO and $HDJ_{NCP}$

[Figure]

Figure R3. The variation in normalized $HDJ_{NCP}$ (orange) and BSISO (green) from 1980 to 2015 after the removal of the linear trend. The "○" indicates the anomalies of $HDJ_{NCP}$ and BSISO with the same mathematical sign. The "●" indicates the anomalies of $HDJ_{NCP}$ and BSISO with significant amplitudes (i.e., |anomalies| > $0.8 \times$ its standard deviation).

**29. Line 109, Do authors have any idea why negative SST anomalies disappear in November?**

*Reply:*

This question was not the mainly concerned issue of this manuscript, but we tried to provide a reasonable guess.

The disappearing of the negative SST anomalies in November connected with the change of the atmospheric circulations and can be explained by Figure 6 in the manuscript. (1) The September-October negative SST anomalies in the west of Beaufort Sea co-varied with the positive sea ice anomalies. (2) According to many previous studies, the signal of the ice in the polar region cannot persistent for long time by itself. **Its influence should delivery via the change of the atmospheric circulations.** (3) In Figure R4 (i.e., Figure 6 in the MS), the distribution and intensity of the atmospheric circulations associated with BSISO was different in September-October and November both near surface and in the mid-troposphere. (4) In November, the local atmospheric circulation associated with positive BSISO was the significant pressure gradient between an anti-cyclonic and a cyclonic circulation (Figure R4), which **weakened the release of the surface heat flux and did not maintain the negative SST anomalies in November.**

[Figure]

Figure R4. The CC between BSISO and September-October (a) geopotential height (shading), wind (arrow) at 500 hPa, (c) SLP (shade), and surface wind (arrow); and November (b) geopotential height (shading), wind (arrow) at 500 hPa, (d) SLP (shade), and surface wind (arrow) from 1979 to 2015, after detrending. The white dots indicate CCs exceeding the 90% confidence level (t test). The black box in (a–d) represents the location of the Beaufort Sea.

**31. Line 118, why authors cannot directly link the SST anomalies over BS and GA to HDJ?**

*Reply:*

Detailed comments 31 was similar with the Major comment 1.

Certainly, there was directly link between the November SST anomalies and the number of haze days in December and January. There are two reasons why we link the number of haze days and the September-October sea ice. (1) As an efficient driver, the September-October sea ice was one month in advance of the November SST, which **supports sufficient time gap to make the seasonal prediction in the real-time operation.** That is, in November, we may gain the sea ice September-October data and run the statistical seasonal prediction models. (2) The goal of this manuscript is to reveal the connection between the sea ice and the haze pollution in the early winter. Our studies **not only reveal the link between the SST and haze, but also deepen the understanding the impacts of the sea ice on the haze** by taking the SST anomalies as a bridge.